# CycloPepper: a machine learning platform for predicting cyclization outcomes and optimizing synthesis of therapeutic cyclopeptides

Yourong Pan[1,2,5], Chengrui Hu[1,2,5], Jiaqi Li[3], Feng Wan[1,2], Xin Hong [3,4] ✉ & Chengxi Li [1,2] ✉

Cyclic peptides exhibit remarkable stability, membrane permeability, and binding affinity, positioning them as promising therapeutics. However, their synthesis, particularly on-resin head-to-tail cyclization, remains challenging, with cyclization site selection critically influencing yield. Here, we introduce a machine learning (ML) approach to predict cyclization outcomes, leveraging CycloBot, our fully automated cyclic peptide synthesis platform. Using this system, we generate a standardized dataset of 306 cyclic peptides (2–14 residues) and develop an ML model achieving an average prediction accuracy of 84%. Experimental validation with 74 random and therapeutic peptides showed an 86% prediction consistency. To facilitate practical use, we built CycloPepper, a user-friendly platform available through both web and software interfaces, enabling rapid cyclization site assessment. This tool effectively identified potential cyclization sites for disease-targeting peptides, including cancer biomarkers. Our work illustrates the potential of ML-assisted synthesis to streamline cyclic peptide synthesis and accelerate therapeutic discovery.

Cyclic peptides represent a structurally diverse class of biomolecules, exhibiting a broad spectrum of shapes, sizes, and chemical functionalities[1]. Their natural abundance underscores their evolutionary significance, with many displaying potent biological activities ranging from hormonal signaling (e.g., oxytocin and vasopressin) to antimicrobial defense (e.g., vancomycin and its clinically approved derivatives telavancin, dalbavancin, and oritavancin)[2–5]. The constrained architecture of cyclic peptides often confers superior pharmacological properties compared to their linear counterparts, including enhanced binding affinity and specificity[6], improved proteolytic stability[7], and increased membrane permeability[2] (Fig. 1a). Additionally, cyclization can refine selectivity and potency, unlocking peptide-based therapeutic potential while enabling differentiated pharmacological profiles[8].

Among the various cyclization strategies, head-to-tail cyclization is particularly advantageous for drug development[9]. Unlike alternative approaches (e.g., side chain-to-side chain, head-to-side chain, or side chain-to-tail cyclization)[10–12], head-to-tail closure eliminates free N- and C-termini, thereby enhancing metabolic stability[2]. Furthermore, this method preserves native amide bonds without requiring transition

[1]Key Laboratory of Biomass Chemical Engineering of Ministry of Education, College of Chemical and Biological Engineering, Zhejiang University, Hangzhou, PR China. [2]Zhejiang Key Laboratory of Intelligent Manufacturing for Functional Chemicals, ZJU-Hangzhou Global Scientific and Technological Innovation Center, Zhejiang University, Hangzhou, PR China. [3]Center of Chemistry for Frontier Technologies, Department of Chemistry, Zhejiang University, Hangzhou, PR China. [4]School of Chemistry and Chemical Engineering, Henan Normal University, Xinxiang, PR China. [5]These authors contributed equally: Yourong Pan, Chengrui Hu. ✉e-mail: hxchem@zju.edu.cn; cxli@zju.edu.cn

**Fig. 1 | Background and cyclization site selection. a** Therapeutic advantages of cyclic peptides over linear peptides. **b** The characteristics of head-to-tail cyclization make it convenient for CAD and prediction. **c** Critical influence of cyclization site selection on successful cyclization. **d** Integrated workflow combining automated synthesis for dataset generation, machine learning for pattern recognition and site prediction, experimental validation, and user-friendly platform development to guide cyclic peptide synthesis.

metal catalysts or special amino acids, ensuring broad compatibility with diverse peptide sequences. The simplicity of this approach also facilitates computational design and optimization[13], enabling the development of cyclic peptides tailored to specific biological targets (Fig. 1b).

Despite these advantages, head-to-tail cyclization remains synthetically challenging. The entropic penalty associated with pre-cyclization conformational organization often hinders intramolecular ring closure, particularly for shorter peptides (<7 residues)[9]. Competing intermolecular reactions, such as dimerization or oligomerization, further complicate the process[9]. Additionally, the inherent preference of amide bonds for trans-configurations promotes extended linear conformations, exacerbating cyclization inefficiencies[9]. Solid-phase peptide synthesis (SPPS) mitigates some purification challenges but introduces steric constraints due to resin tethering, rendering cyclization highly sequence-dependent. For instance, bulky residues near the cyclization site can impede productive ring closure, necessitating careful retrosynthetic disconnection planning (Fig. 1c).

Advances in machine learning (ML) have revolutionized the identification of complex structure-activity relationships, offering insights beyond traditional empirical approaches. In synthetic chemistry, ML-integrated automation has accelerated drug discovery by optimizing reaction pathways and predicting molecular properties[14,15]. Within peptide science, ML has been applied to forecast self-assembly[16], membrane permeability[17], bioactivity (e.g., antimicrobial[18,19], anticancer[20,21], and cell-penetrating peptides[20]), structure prediction and design[22–24]. However, its utility in cyclic peptide synthesis, particularly on-resin head-to-tail cyclization, remains underexplored. A major limitation is data heterogeneity[25]: literature-reported cyclization yields often derive from disparate methods, conditions, and analytical standards, complicating reproducibility and model training[26]. Automated synthesis platforms address this by

generating standardized, high-fidelity datasets[27], as exemplified by Pentelute's high-throughput flow synthesis of small proteins[28,29].

Here, we bridge this gap by combining automated flow synthesis with ML-driven predictive modeling. Leveraging a diaminonicotinic acid (DAN) linker system that enables rapid, cleavage-free on-resin cyclization, we construct a dataset of 306 cyclic peptides (2–14 residues) using the CycloBot automated platform and AutoXpert control software[30]. Twelve ML algorithms are evaluated, yielding an optimized model with 84% accuracy in predicting cyclization feasibility. Experimental validation confirms an 86% success rate, demonstrating good generalization within the scope of the available dataset. To democratize access, we develop CycloPepper, an open-source platform featuring standalone software and a web interface for real-time cyclization prediction. This tool streamlines cyclic peptide design for therapeutic targets, as validated by its performance on the synthesis of disease-relevant peptides. By unifying automated synthesis, ML prediction, and a user-friendly interface, this platform reduces empirical optimization burdens and accelerates the development of functional cyclic peptides (Fig. 1d).

## Results

### Dataset building

To establish a reliable dataset for cyclic peptide synthesis, we employed an on-resin head-to-tail cyclization strategy using DAN resin[30] (Fig. 2a). Linear peptides were first synthesized via standard solid-phase peptide synthesis (SPPS) on DAN resin. Subsequent treatment with isoamyl nitrite facilitated the conversion of the DAN linker into reactive triazolopyridine intermediates, which, upon addition of $N$, $N$-diisopropylethylamine (DIEA) as a base, underwent efficient intramolecular cyclization to afford the desired cyclic peptides. To ensure rapid and reproducible dataset generation, we leveraged an

automated flow synthesis platform, CycloBot[30], which enabled high-temperature on-resin cyclization in a controlled and scalable manner. The system was equipped with precision switching valves, pumps, a temperature-regulated reactor, an in-line UV detector, and dedicated reagent reservoirs (see Supplementary Information Section 2.1 for more details). Reaction mixtures were collected directly from the system and analyzed by liquid chromatography-mass spectrometry (LC-MS). Successful cyclization was determined by the presence of the target molecular weight, with samples assigned a binary classification ("1" for successful cyclization, "0" for unsuccessful cyclization) (Fig. 2b). This fully automated approach not only optimized labor efficiency but also ensured high data consistency, enabling the rapid assembly of a high-quality dataset for downstream machine learning applications.

Given our focus on predicting cyclization feasibility, we established a training dataset of 306 linear peptide sequences (2–14 residues), each corresponding to a single experimentally validated cyclization outcome. Crucially, each entry represents only the specified linear precursor. For example, "FPGM" exclusively denotes cyclization of linear FPGM to cyclo-FPGM, excluding permutation variants such as PGMF. The dataset comprises 174 cyclization-viable sequences (labeled "1") and 132 non-viable sequences (labeled "0"), ensuring balanced representation without outcome bias. Notably, shorter cyclic peptides (2- and 3-residue) exhibited distinct cyclization behaviors due to their constrained backbones, prompting us to prioritize these subsets for deeper analysis. The final stratified dataset comprised 45 two-residue, 53 three-residue, 120 four-to-six-residue, 69 seven-to-nine-residue, and 19 ten-residue and longer cyclic peptides. This stratified distribution facilitated more balanced model training and provided preliminary evidence of the model's generalization capability.

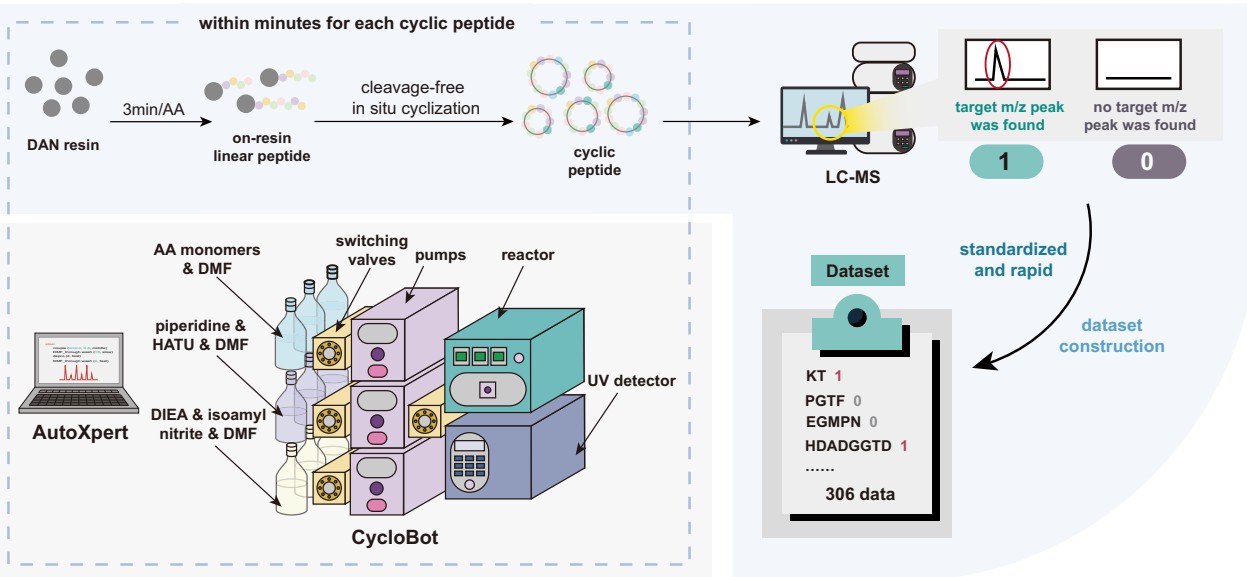

**Fig. 2 | Cyclic peptide dataset construction. a** Synthetic route of on-resin head-to-tail cyclization utilizing DAN resin. **b** Overview of the fully automated dataset construction process. The automated synthesis is carried out by the AutoXpert control software and CycloBot automated synthesis system. Cyclic peptide products were detected by LC-MS.

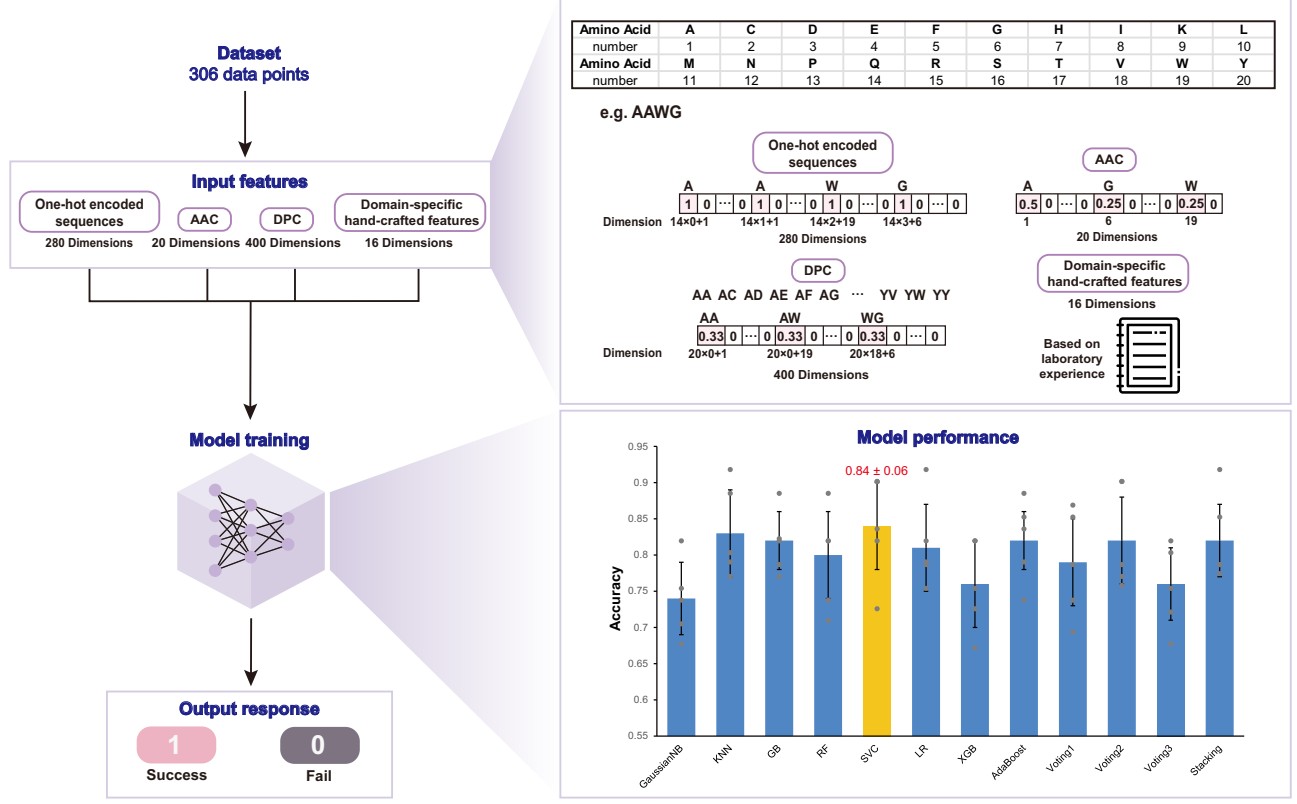

**Fig. 3 | Overview of machine learning models.** Input features include one-hot encoded sequences, AAC, DPC, and domain-specific hand-crafted features. The principles of each encoding are illustrated in the figure. Model selection is based on the average accuracy values. Gray dots represent individual accuracy data points from cross-validation folds, and bar heights indicate the mean accuracy. Error bars represent standard deviation ($n = 5$ folds), and data are presented as mean values ± SD. The GaussianNB model shows the lowest mean accuracy (0.74), whereas the SVC model shows the highest mean accuracy (0.84), with its bar highlighted in yellow. For additional model performance details, see Supplementary Table 4.

## Peptide sequences encoding and feature engineering

To enable reliable analysis of complex biological and chemical data and achieve high predictive accuracy, we established a systematic methodology for peptide sequence encoding and feature extraction. Our approach leveraged sequence-based encodings to capture essential structural and functional attributes of cyclic peptides (Fig. 3).

One-hot encoded sequences: Each peptide sequence, derived from cyclic peptides by cleavage at cyclization sites, was encoded as a fixed-length numerical vector to facilitate machine learning analysis. We employed one-hot encoding to represent each amino acid in the peptide sequence as a 20-dimensional binary vector, where each position corresponds to one of the 20 canonical amino acids. For instance, alanine (A) is encoded as [1, 0, 0,..., 0], with the "1" in the position corresponding to alanine. Given a maximum sequence length of 14 residues, each sequence was represented as a concatenation of fourteen 20-dimensional vectors. Sequences shorter than 14 residues were padded with zero vectors to maintain uniform dimensionality, resulting in a final 280-dimensional feature vector per sequence.

Amino acid composition (AAC) measures the normalized frequency of the 20 standard amino acids in a peptide sequence, producing 20-dimensional feature vectors. It is computed as:

$$f(a) = \frac{N(a)}{L}, a \in \{A, C, D, \ldots, Y\} \tag{1}$$

Here, $f(a)$ represents the frequency of amino acid $a$, $N(a)$ is the count of amino acid $a$ in the peptide sequence, and $L$ is the total length of the sequence.

Dipeptide composition (DPC) generates a 400-dimensional feature vector, calculated as:

$$D(r, s) = \frac{N_{rs}}{L - 1}, r, s \in \{A, C, D, \ldots, Y\} \tag{2}$$

In this case, $N_{rs}$ refers to the number of dipeptides composed of amino acids $r$ and $s$.

Domain-specific hand-crafted features: In addition to sequence-based encodings, we designed a set of 16-dimensional hand-crafted features derived from empirical observations of peptide cyclization efficiency. These features were formulated through systematic analysis of synthetic outcomes, incorporating domain knowledge to enhance predictive performance. For example, inductive reasoning from experimental data revealed that sequences terminating in proline (P) or cysteine (C) exhibit reduced cyclization propensity, likely due to steric constraints or unfavorable conformational dynamics. Conversely, glycine (G) at the $N$-terminus was found to promote cyclization, presumably owing to its minimal steric hindrance and conformational flexibility. To encode these trends, binary indicators (1/0) were assigned to sequences exhibiting these features, thereby integrating empirical synthetic knowledge into the feature space.

Collectively, this multi-faceted feature engineering strategy, combining one-hot encoding, AAC/DPC descriptors, and hand-crafted experimental insights, provided a solid foundation for predictive modeling of cyclic peptide behavior. The resulting 716-dimensional feature vectors (280 from one-hot encoding, 20 from

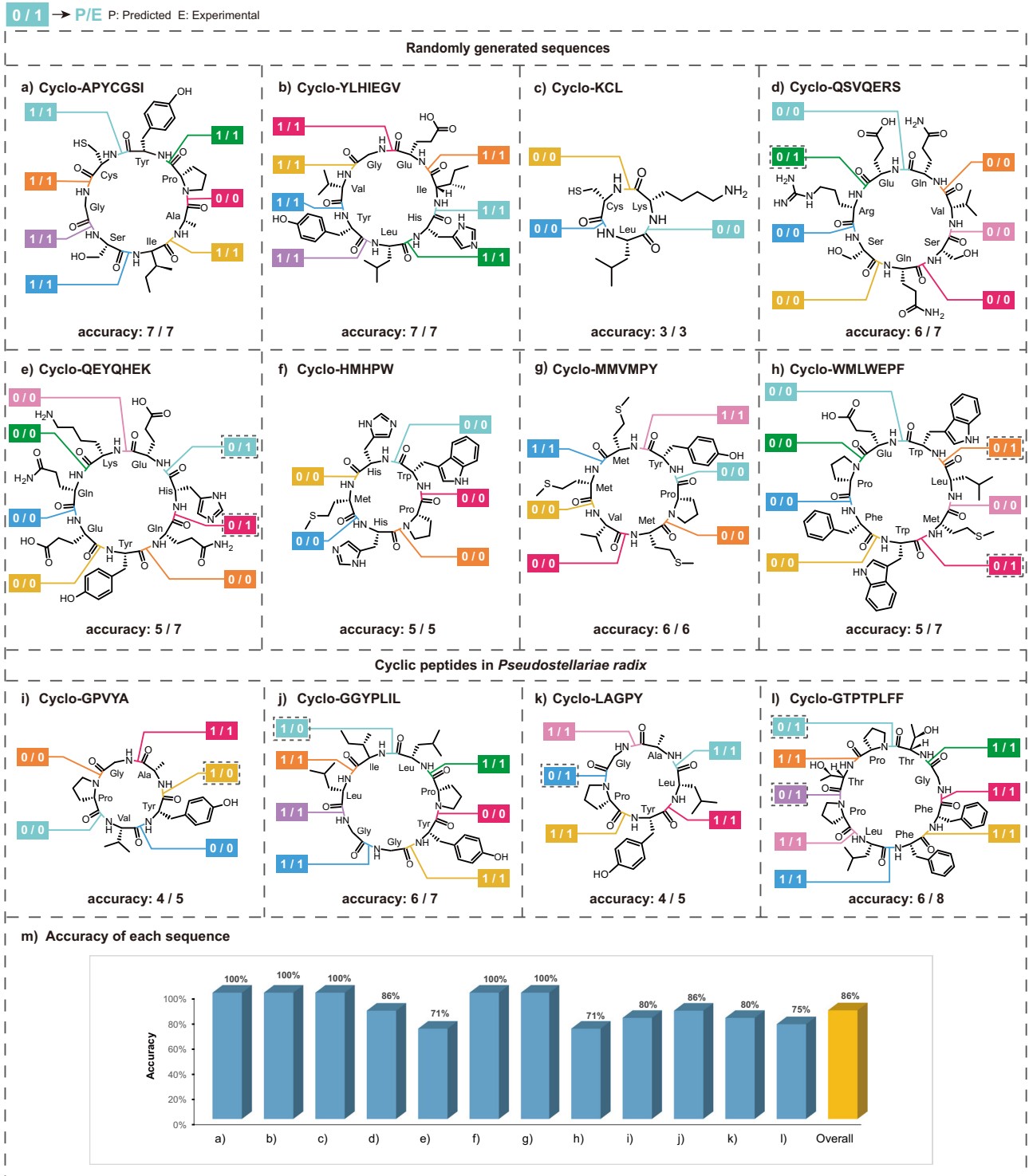

**Fig. 4 | Agreement between predicted cyclization outcomes and experimental validation.** Cyclization sites are indicated by colored amide bonds, and results are displayed within colored boxes as P/E, where P denotes the predicted cyclization outcome (1 = feasible, 0 = infeasible) and E denotes the experimental result (1 = successful, 0 = not observed). Boxes outlined with a gray dashed border indicate discrepancies between prediction and experiment, whereas boxes without a dashed border indicate agreement between the two. Panels **a**–**h** show randomly generated cyclic peptides. Panels **i**–**l** show cyclic peptides derived from *Pseudostellariae radix* exhibiting tyrosinase inhibitory activity. Panel **m** summarizes the accuracy of individual cyclic peptides and the overall prediction accuracy, with the latter highlighted by a yellow bar.

AAC, 400 from DPC, and 16 from hand-crafted features) ensured a comprehensive representation of sequence information, laying the foundation for accurate and interpretable machine learning predictions.

## ML model selection and performance evaluation
Before constructing the machine learning model, we first concatenated the derived feature vector, followed by the removal of all zero dimensions. The processed dataset was then employed as input

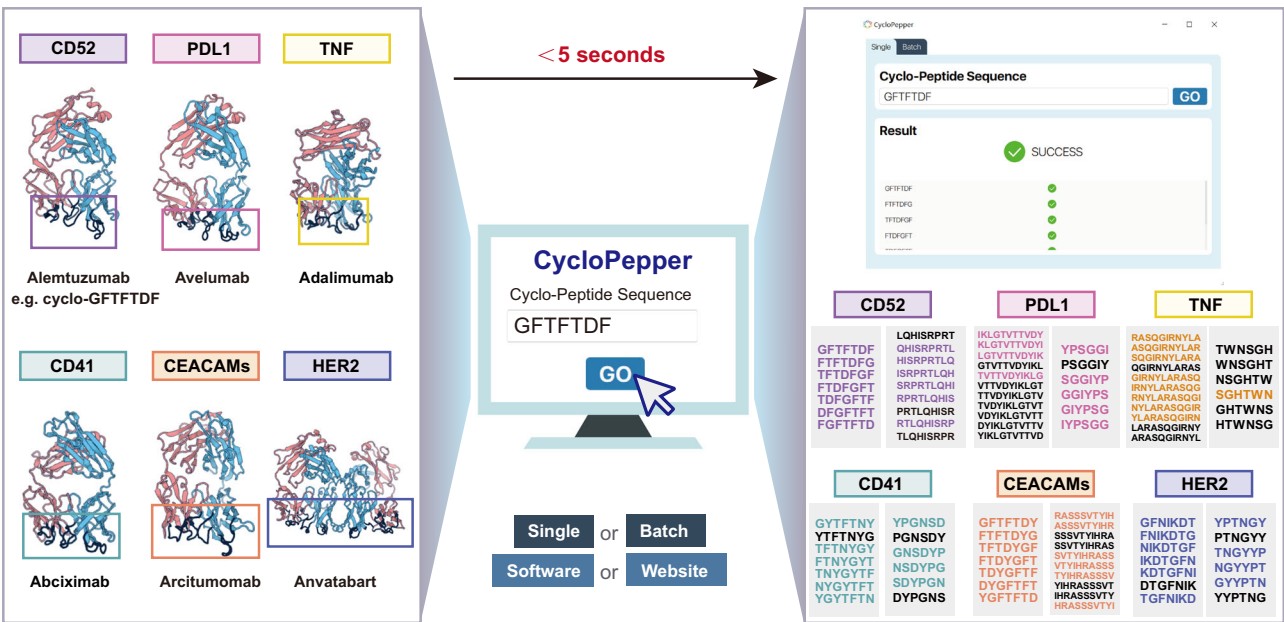

**Fig. 5 | A user-friendly platform for rational design of bioactive cyclic peptides.** The CycloPepper enables rapid evaluation of cyclization feasibility for both individual and multiple peptide sequences. Six antibodies and their complementarity-determining regions are presented, with binding sites highlighted in dark blue and outlined with colored boxes. Users submit peptide sequences and obtain real-time predictions of successful cyclization sites (color-coded for viable sites versus black for non-cyclizing positions), providing an efficient tool for optimizing synthetic routes to therapeutic cyclic peptides targeting diverse diseases. For more model performance details, see Supplementary Table 6.

for model training and evaluation. In our systematic search for the optimal predictive algorithm, we assessed the performance of twelve distinct machine learning models, encompassing both individual classifiers and ensemble methods Fig. 3. The evaluated algorithms included Gaussian Naive Bayes (GaussianNB), k-Nearest Neighbors (KNN), Gradient Boosting Classifier (GB), Random Forest Classifier (RF), Support Vector Classifier (SVC), Logistic Regression (LR), Extreme Gradient Boosting (XGB), Adaptive Boosting Classifier (AdaBoost), and four ensemble configurations: (i) a voting classifier combining GaussianNB, RF, SVC, and XGB (Voting1), (ii) a voting classifier integrating RF and SVC (Voting2), (iii) a voting classifier incorporating GaussianNB, RF, and SVC (Voting3), and (iv) a stacked classifier composed of GaussianNB, RF, and SVC (Stacking).

Model performance was evaluated using five-fold cross-validation to ensure statistical reliability. The results indicated that the differences among the learning algorithms were not statistically significant, as all performances fell within one standard deviation. The GaussianNB classifier displayed the lowest performance ($0.74 \pm 0.05$). Several models, including RF ($0.79 \pm 0.06$), LR ($0.81 \pm 0.07$), and XGB ($0.76 \pm 0.06$), achieved moderate accuracies within the 0.70-0.80 range. Slightly higher accuracies were obtained for GB ($0.82 \pm 0.04$), AdaBoost ($0.82 \pm 0.05$), and the stacking ($0.82 \pm 0.06$). Notably, KNN ($0.83 \pm 0.06$) and SVC ($0.84 \pm 0.06$) delivered the highest mean accuracies. Given the close performance and comparable variance, the SVC model with the highest mean accuracy was selected for subsequent experimental validation and predictive applications.

### Experimental validation of ML model
To assess the predictive performance of our ML model, we implemented a validation protocol wherein each candidate cyclic peptide sequence was systematically evaluated across all potential cyclization sites. For the 8 randomly generated sequences (Fig. 4a–h): five 7-residue sequences, cyclo-APYCGSI, cyclo-YLHIEGV, cyclo-QSVQERS, cyclo-QEYQHEK, cyclo-WMLWEPF, a

6-residue sequence, cyclo-MMVMPY, a 5-residue sequence, cyclo-HMHPW, and a 3-residue sequence, cyclo-KCL, we synthesized every linear precursor by changing cyclization sites. Specifically, cyclo-KCL yielded three linear precursors (KCL, CLK, LKC), while each 7-residue peptide generated seven precursors. Following automated solid-phase synthesis using DAN resin under standard conditions, reaction outcomes were verified via LC-MS detection of the desired cyclic products. The experimental results demonstrated overall agreement with the model's predictions, with cyclization accuracies of 71% for both cyclo-QEYQHEK and cyclo-WMLWEPF, 86% for cyclo-QSVQERS, and 100% for cyclo-APYCGSI, cyclo-YLHIEGV, cyclo-KCL, cyclo-HMHPW, and cyclo-MMVMPY. These findings underscore the model's reliability in identifying viable cyclization sites within the evaluated dataset.

To further validate the model's practical utility, we extended our investigation to four biologically active cyclic peptides derived from *Pseudostellariae radix*[31], including *Heterophyllin J* (5-residue), *Pseudostellarins D* (7-residue), *Pseudostellarins A* (5-residue), and *Pseudostellarins H* (8-residue) (Fig. 4i–l). Comparative analysis between the predicted and experimentally observed cyclization efficiencies revealed a substantial degree of consistency, with prediction accuracies of 80% for *Heterophyllin J*, 86% for *Pseudostellarins D*, 80% for *Pseudostellarins A*, and 75% for *Pseudostellarins H*.

In the entire experimental validation set, 39 sequences were cyclizable, while 35 were non-cyclizable, providing a nearly balanced dataset for evaluating model performance. After conducting 74 synthesis experiments across seven distinct cyclic peptide sequences, 64 predictions were consistent with experimental results. The aggregate predictive accuracy of the model reached 86% (Fig. 4m), supporting its reliability in guiding the selection of suitable cyclization sites for peptide synthesis within the evaluated dataset. This success rate suggests the model could serve as a useful tool for synthetic chemists, assisting in the design and optimization of cyclic peptides while minimizing experimental trial and error.

### CycloPepper prediction for bioactive cyclic peptides

To address the critical challenge of translating computational predictions into practical cyclic peptide synthesis, we developed CycloPepper, a platform incorporating our optimized machine learning algorithm (Fig. 5). This versatile tool, accessible through both standalone software and web interface, enabled: (1) single-sequence analysis (e.g., GFTDFGF), (2) batch processing of multiple sequences via CSV upload (software only), and (3) real-time prediction of cyclization feasibility across all potential sites. The web interface is accessible from mobile devices, allowing researchers to easily perform predictions from their phones, enabling seamless predictions anytime and anywhere. The platform's automated workflow was initiated upon activation of the execution module, wherein input sequences were transformed into 716-dimensional feature vectors, processed by our SVC model, and yielded results within seconds. The output interface provides immediate quality assessment, displaying "SUCCESS" for sequences containing ≥1 viable cyclization site, with detailed positional annotations (green checkmarks indicating successful sites versus red crosses for non-viable positions) to guide synthetic efforts.

We validated CycloPepper's utility across seven therapeutically relevant protein targets: CD52, PD-L1, TNF-α, CD41, CEACAMs, HER2, and APP, leveraging known antibody complementarity-determining regions as design templates[32]. Systematic analysis of 561 designed cyclic peptides revealed 341 (60.8%) feasible cyclization sites, with some sequences only exhibiting 1–2 viable cyclization sites. This predictive capability offered three key advantages for peptide drug development: elimination of 39.2% non-viable sequences prior to synthesis, accelerated identification of bioactive candidates, and substantial reduction in resource expenditure, decreasing both synthetic timelines and associated costs by enabling data-driven sequence prioritization.

## Discussion

This study established an integrated experimental-computational framework for predictive cyclic peptide design. By employing automated solid-phase peptide synthesis with DAN resin, we generated a high-quality training dataset encompassing diverse on-resin head-to-tail cyclization outcomes. Through systematic machine learning evaluation, we identified an SVC model that achieved an average cross-validation accuracy of 84% in cyclization site prediction, with subsequent experimental validation showing 86% prediction consistency. The implementation of this optimized algorithm in our CycloPepper platform enables real-time analysis of individual peptide sequences, high-throughput batch processing capabilities, and reliable identification of potential cyclization motifs across multiple therapeutic targets. Notably, CycloPepper offers user-friendly convenience, as it can be accessed both via software on a computer and through a mobile-friendly website, allowing users to quickly and easily assess cyclization sites from anywhere. This work provides researchers with a practical computational tool that bridges the gap between sequence design and synthetic feasibility, accelerating the development of bioactive cyclic peptides for pharmaceutical applications. The combination of reliable predictive modeling and user-friendly accessibility highlights CycloPepper's potential as a valuable resource in peptide-based drug discovery.

## Methods

### Rapid cyclic peptides dataset building

Comprising 306 linear peptides, the dataset was engineered to ensure comprehensive representation of all 20 native amino acids, with each type appearing in more than 20 distinct sequences to support balanced model training and reliable evaluation. The seed sequences were randomly generated using a programmatic approach to ensure the inclusion of all 20 native amino acids. From this pool, specific subsequences were selected for the directed variation strategies,

including residue addition (e.g., TDDA to TDDAG), deletion, and substitution. These modifications were manually performed (Supplementary Information Section 2.5). Minimal sequence alterations revealed critical structure-cyclability relationships while maintaining broad heterogeneity without redundancy, as evidenced by TDDA (no cyclization) versus TDDAG/GTDDA (successful cyclization).

The main step of dataset construction is the synthesis of cyclic peptides. The first step is manual DAN resin synthesis. Forty milligrams of DAN resin (0.36 mmol/g loading) were used in all experiments in the dataset. The next step is automated synthesis. All cyclic peptides were synthesized on the automated instrument CycloBot along with control software AutoXpert. The following stock solutions were used for cyclic peptide synthesis: Fmoc-protected amino acids, including 20 native amino acids. Activating agent *O*-(7-Azabenzotriazol−1-yl)-*N*, *N*, *N′*, *N′*-tetramethyluronium hexafluorophosphate (HATU) as a 0.38 M stock solution in DMF, DIEA (10% v/v in DMF), DIEA (1% v/v in DMF), and deprotection stock solution (40% piperidine and 1% HCOOH (v/v) in DMF, which was diluted inline to 20%(v/v) and 0.5%(v/v) during synthesis). The automated process includes linear peptide synthesis at 90 °C, activation at room temperature (25 °C), and cyclization at 50 °C. The reaction mixtures were characterized by LC-MS (for more details, see Supplementary Information Section 2).

### Model training

We made a 716-dimensional vector feature depending on the cyclic peptide sequences, which consists of one-hot encoded sequences, amino acid composition (AAC), dipeptide composition (DPC), and domain-specific hand-crafted features. After removing all 0 dimensions, it ended up with a 414-dimensional vector (for more details, see Supplementary Information Section 3).

### Featurization

Twelve ML model architectures were implemented in the Python programming environment, version 3.7.1. Five-fold cross-validation was used for evaluating model performance. ML algorithms were mostly imported from the module "scikit-learn".

### Similarity evaluation

Following the DOME recommendations for supervised machine learning validation in biology[33], we measured sequence similarity as percent identity, quantifying the independence between training set and experimental validation set by computing global pairwise identity between each experimental amino-acid sequence and all training sequences on the linearized peptides. Since cyclic peptides can be represented with different starting positions when linearized, we performed a rotation-invariant audit for high-similarity pairs (initial identity ≥60%) (for more details, see Supplementary Information Section 4.2).

### Reporting summary

Further information on research design is available in the Nature Portfolio Reporting Summary linked to this article.

## Data availability

All the relevant data generated in this study are provided in the Supplementary Information and Source Data. The training dataset of 306 cyclic peptides, along with all prediction results, has been deposited in the GitHub repository under https://github.com/Yongboxiao/Selection-of-Cyclization-Sites. Source data are provided with this paper.

## Code availability

The Python code for the machine learning model described in this manuscript is available in a GitHub repository (https://github.com/Yongboxiao/Selection-of-Cyclization-Sites) and archived on https://

doi.org/10.5281/zenodo.18154994[34]. Additionally, the CycloPepper platform can be accessed through its user-friendly website interface at http://www.cyclopepper.com/.

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

## Acknowledgements

This work was financially supported by the National Natural Science Foundation of China (Grant Nos. 22208290, C.L.; 22122109, X.H.; 22271253, X.H.; W2512004, X.H.). The China Postdoctoral Science Foundation (Grant No. 2023M743085, F.W.). National Key R&D Program of China (Grant No. 2022YFA1504301, X.H.). New Generation Artificial Intelligence-National Science and Technology Major Project (Grant No. 2025ZD0121905 X.H.).

## Author contributions

Y.P. contributed to the main idea, data collection, algorithm design, development of machine learning code, and manuscript writing. C.H. was involved in optimizing the machine learning algorithms, developing the software and website, and writing relevant sections of the manuscript. J.L. contributed to the algorithm design and provided valuable suggestions. F.W. assisted with data collection and experimental validation. X.H. and C.L. supervised the research and participated in the manuscript revision, and were responsible for the overall quality and direction of the work.

## Competing interests

The authors declare no competing interests.
