## [Transparent Peer Review file · Nature Communications]

CycloPepper: A Machine Learning Platform for Predicting Cyclization Outcomes and Optimizing Synthesis of Therapeutic Cyclopeptides

Corresponding Author: Dr Chengxi Li

Version 0:

Reviewer comments:

Reviewer #1

(Remarks to the Author)

Summary

The authors present a ML approach for predicting peptide viability for head-to-tail cyclization. The authors constructed a dataset with 306 cyclic peptides (2-14 residues) using their own automated flow synthesis protocol. This dataset was stratified depending on peptide size into di-, tri-, tetra- to hexa-, hepta- to nona- and decapeptides or longer.

Strengths

The webserver interface is intuitive and easy to use.

The purpose of this tool is interesting and would be valuable for peptide design.

The experimental validation, although limited in its scope, is good for demonstrating the practical utility of the tool.

Weaknesses

(1) In the results section there is no mention to the model performance being measured in a 70:30 train/test random split. It gives the impression that the results are from the five-fold cross-validation. In fact, considering that the model selection is done looking at the train/test split, the five-fold cross validation seems redundant. It would have been better to not do the train/test split and instead perform model selection and report the results in the 5-fold cross validation with the associated standard deviation.

(2) Experimental validation procedure is unclear as to what the percentage of model accuracy corresponds to. I assume from the figure that for each sequence, several experiments were performed in which the N- and C-termini were moved. For example, for cyclo-KCL, the sequence KCL, CLK, and LKC were predicted with the model and experimentally validated. I would appreciate it if the authors could clarify this point both in their response and in the manuscript.

(3) It was not clear in the initial training dataset if the training peptides were synthesised as all possible linear starts/stops as illustrated in results for Fig 4. Does 306 peptides mean 306 unique peptides with all their different permutations (so more than 306 sequences in the actual dataset) or 306 unique sequences (which are permutations of each other)? The larger peptides would be much more over-represented in training if the former is true.

(4) Figure 4 legend should specify what the symbols in colour 1/0 means, does this mean predicted to be cyclised, but not observed experimentally, or the other way around?

(5) Was the feature selection process repeated for each cross-validation fold, or for the 70% of the data in the training set, or for the whole dataset. The authors should clarify in their response and the manuscript. If the feature selection was performed in the whole dataset, that could introduce overfitting, it would have been more appropriate to perform it only in the training data or the cross-validation folds.

(6) The process for generating the random sequences should be specified. It is also important that the authors report how similar these sequences are to the sequences in their training data. If they are too similar, then the authors cannot claim robust generalization.

(7) The link to the code repository is not publicly available. The authors should make the code publicly available that is used to train the models and generate the sequences, to verify the claims made in the text.

(8) The authors should make available the training and evaluation data in order to assess claims made.

(9) The authors do not specify what proportion of the peptides were positive and which were not in the training dataset.

(Remarks on code availability)

Code was not available.

Reviewer #2

(Remarks to the Author)

(Remarks on code availability)

Storing the model as a pickle file, makes it more difficult to reuse it, as the pickled file is environment specific. Otherwise the code seems reproducible enough.

Reviewer #3

(Remarks to the Author)

A standard way to improve the therapeutic competence of peptides is to cyclize them. This can be done in many ways, including sidechain-to-sidechain, head-to-sidechain, sidechain-to-tail, and head-to-tail. Head-to-tail cyclization presents several advantages. It eliminates free N- and C-termini, retains sidechain stereochemistries, and is less affected by steric constraints. Yet not all peptides readily cyclize, and there can be many factors that complicate head-to-tail unions. For instance: entropic penalties, competitive dimerization, and limited flexibility. Just because a peptide could cyclize doesn't mean it will, and solid-phase synthesis can only offset some of the synthetic challenges. Li et al aim to remove the guesswork of whether a peptide will cyclize or not by developing a predictive algorithm. In brief, Li et al developed an automated synthesis platform to generate a library (306 members; size range 2-14mer) of cyclic head-to-tail peptides using DAN resin technologies. Then, they deployed this library to train a machine learning model to predict cyclization efficiencies (both head-to-tail and across all other sites) in linear peptides using one-hot encoding, AAC/DPC descriptors, and hand-crafted experimental insights. What resulted is an SVC model with 86% cross-validation accuracy in cyclization site prediction and 83% success rate. Predictive accuracy was verified by modeling three randomly generated cyclic peptide sequences and four biologically active cyclic peptides derived from *Pseufostellariae Radix*. The model was made available via an open-source platform known as "CycloPepper" with standalone software and a web interface.

Overall, the CycloPepper software is a significant advance in the field, and this reviewer suggests that the manuscript be considered after revisions. The following points should be addressed.

(1) Although the distribution in peptide size is well presented, the authors should include the amino acid diversity score of the starting library. In other words, how similar (or different) are the starting sequences in terms of amino acid similarity/identity and three-dimensional structure? How were the 306 peptides selected?

(2) In assigning the binary classification, "1" signifies that the target molecular weight was observed and "0" if it was not. It is unclear if the target molecular weight refers exclusively to the parent ion and what degree of precision was considered for successful identification. The authors should provide these details. Looking at the Supporting Information, it appears that $[M+H]^+$ and $[M+Na]^+$ are being considered. The authors should also state if they attempted to validate successful cyclizations by checking for adducted masses or smaller fragments.

(3) One limitation of the training model is that it only accounts for the twenty native amino acids. It does not account for non-proteinogenic residues, which are frequently included in peptides to improve drug-like properties as well as binding affinity and target selectivity. It would be good to consider peptides harboring some of the most frequently encountered non-proteinogenic residues in the model.

(4) In the Supporting Information, the 'Found: $[M+H]^+$ ' for cyclo-YLHIEGV is incorrect. What is written does not match the provided MS spectra.

(5) In the Supporting Information, some of the peptides form $[2M+Na]^+$ in addition to $[M+H]^+$. As outlined in the successful cyclization criteria, this would still be considered successful because it made some of the correct product in addition to the dimer. Can the model predict oligomerization based on the current training data? Aside from cyclization, the risk of oligomerization would be an important parameter for peptide medicinal chemists designing sequences.

(Remarks on code availability)

I reviewed the code, but I am not an expert. I did not install and run the code.

Version 1:

Reviewer comments:

Reviewer #1

(Remarks to the Author)

We appreciate the effort the authors have made to address the concerns from the reviewers and improve the paper.

Main issue

However, there remains one main problem with the paper: in its current state the experimental validation is insufficient (only 7 peptides are used as a hold out test set). Moreover, given the enrichment in cyclization sites in the evaluation set (there are 32 positive cyclization sites against 10 negative ones), the expected model accuracy for a model that always predicts "1" would be 76.2% with CycloPepper showing an accuracy of 83%, is only 8% above. It would be more appropriate to report binary classification metrics robust to class imbalance (like Matthew's correlation coefficient). Alternatively, the discussion of the results should clearly highlight that the expected baseline accuracy for a classifier on this dataset is 76.2% to properly contextualize the results.

Other issues arising from further inspection

After a closer examination of Figure 3 and the code, the model seems to be always predicting peptide cyclization sites where the N belongs to Pro to be negative, even in some cases where it is positive. I would recommend an ablation study to see whether the combination of features used in the study is necessary or positive, or whether the one-hot encoding is sufficient to make the predictions.

Reading Comment_9, the proportion of labels in the training dataset is 56.9%, which is our closest estimate to the expected probability of a randomly sampled peptide to be cyclizable. However, the expected probability looking only at the evaluation set is 76.2%. This highlights the problem with the sample size in the evaluation set, which does not allow for even getting appropriate estimators for the expected probability of cyclization sites.

Finally, after reviewing the code, there is an additional problem with the approach, the MinMaxScaler that is applied to `X_train` in the code block that starts with "#Concatentation" should not be fitted to the whole training dataset, it should be applied to each training fold individually as it is part of model building and feature selection.

We clarify below some minor points that are still unsolved. These issues are fixable and would only require minor comments.

Regarding Comment_1:

The clarification of the method of evaluation is clear. We continue recommending that the standard deviation or standard error of the mean across the 5 folds is included when reporting model performance, not only the average. The purpose of this recommendation is to give an intuition to readers of how significant are the differences between different approaches considered in the study. For example, this would affect lines 172, 196, or 199. Similarly, figure 3 should also display error bars representing that deviation.

This is the bare minimum for Machine learning model reporting (see [1] for a discussion on more robust statistics methods for comparing machine learning models).

Regarding Comment_6:

The similarity between the sequences in training and testing should be reported (see [2]) to give a sense of how generalizable the approach really is. This includes both random sequences and the sequences from *Pseudostellariae radix* (incidentally, in Figure 4, there is a typo in the species name which should be italicised and the second name in lower case).

Regarding Comment_1 to Reviewer #3:

Authors should include further details on what the seed sequences are. Additionally, if the directed variation strategies were performed according to an algorithm it should be clearly stated and the text should direct to the algorithm in the supplementary; alternatively, it should be clearly stated that this process was manually performed.

References:

1. Ash JR, Wognum C, Rodríguez-Pérez R, Aldeghi M, Cheng AC, Clevert D-A, et al. Practically significant method comparison protocols for machine learning in small molecule drug discovery. ChemRxiv. 2024; doi:10.26434/chemrxiv-2024-6dbwv-v2 This content is a preprint and has not been peer-reviewed.

2. Walsh I, Fishman D, Garcia-Gasulla D, Titma T, Pollastri G, Harrow J, Psomopoulos FE, Tosatto SC. DOME: recommendations for supervised machine learning validation in biology. Nature methods. 2021 Oct;18(10):1122-7.

(Remarks on code availability)

There is an a problem with the approach, the MinMaxScaler that is applied to X_train in the code block that starts with "#Concatentation" should not be fitted to the whole training dataset, it should be applied to each training fold individually as it is part of model building and feature selection.

Reviewer #2

(Remarks to the Author)

(Remarks on code availability)

The code is fine. Notebooks are not the greatest way to obtain reproducibility, but the code is clear and can be easily understood. Models are saved as pickled files, which are not the optimal way to obtain interoperability (preferred would be ONNX), but they provide details about specific versioning.

There is an incongruity between the code and the text. The text claims that the feature selection is performed within each training dataset, however, the MinMaxScaler that is applied to X_train in the code block that starts with "#Concatentation" should be fitted to the whole training dataset and then a feature filtering step is applied. This is effectively an additional feature selection step.

Reviewer #3

(Remarks to the Author)

This reviewer thanks the authors for fully addressing the prior critiques. The manuscript is now acceptable for publication.

(Remarks on code availability)

Version 2:

Reviewer comments:

Reviewer #1

(Remarks to the Author)

The authors have addressed most of our concerns. There is still the issue of the dataset and evaluation sets being too small for the strong claim of a "robust generalization", we do not see anything in the paper to justify that claim. Otherwise, we think the paper can be accepted, provided that the following two issues are addressed:

- Comment 5: We appreciate the inclusion of standard deviation to the results. With regards to the analysis, it is clear from the standard deviation that there is no significant effect of the choice of learning algorithm (all of them are within 1 std), the way in which the results are expressed seems to suggest the opposite. It is reasonable to choose the model with the highest average performance.

- Comment 7: Fivefold cross validation cannot be used with a random split of 70% training and 30% test; fivefold means there are five equally sized folds each of them with 20% of the data ($100/5=20$), training is performed in 4 folds and testing on the fifth, in that way all folds are used for testing once. Did you perform fivefold cross-validation or repeated the experiment five times with 70/30 random split? Both things are mutually exclusive.

(Remarks on code availability)

Reviewer #2

(Remarks to the Author)

(Remarks on code availability)

See previous review.

Dear Reviewers,

We sincerely appreciate your rigorous evaluation and insightful critiques of our manuscript (ID: NCOMMS-25-33268). Your expertise and meticulous attention to detail have been instrumental in elevating the scientific quality of this work.

In response to your valuable recommendations, we have undertaken extensive revisions with utmost diligence. Each comment has been addressed through substantive modifications to the manuscript text, supplementary materials, and data repositories. Detailed point-by-point responses follow below, with reviewer comments in italics and enumerated concerns for clarity. Revised content is explicitly highlighted in blue throughout the updated documents.

Reviewer #1 and **Reviewer #2**

Comment_1:

“In the results section there is no mention to the model performance being measured in a 70:30 train/test random split. It gives the impression that the results are from the five-fold cross-validation. In fact, considering that the model selection is done looking at the train/test split, the five-fold cross validation seems redundant. It would have been better to not do the train/test split and instead perform model selection and report the results in the 5-fold cross validation with the associated standard deviation.”

Our response:

We acknowledge the reviewer’s perceptive comment regarding the potential ambiguity in reporting the performance metrics. The reviewer rightly pointed out the need for clarity in distinguishing between evaluation methodologies. In response, we have carefully revised the manuscript to explicitly state that the reported performance metrics (e.g., 86% accuracy) were obtained exclusively through 5-fold cross-validation, rather than a fixed train/test split. This clarification has been revised in the “**Methods**” section of the **manuscript**, which addresses the previously unclear descriptions, ensuring a more precise and unambiguous explanation of the machine learning process. We thank the reviewer for bringing this important matter to our attention.

Before:

Featurization

Twelve ML model architectures were implemented in the Python programming environment, version 3.7.1. Fivefold cross-validation was used for a random split of 70% training, and 30% held-out testing datasets. ML algorithms were mostly imported from the module “scikit-learn”.

After:

Featurization

Twelve ML model architectures were implemented in the Python programming environment, version 3.7.1. ML algorithms were mostly imported from the module “scikit-learn”. **Model performance was evaluated via 5-fold cross-validation.**

Comment_2:

“Experimental validation procedure is unclear as to what the percentage of model accuracy corresponds to. I assume from the figure that for each sequence, several experiments were performed in which the N- and C-termini were moved. For example, for cyclo-KCL, the sequence KCL, CLK, and LKC were predicted with the model and experimentally validated. I would appreciate it if the authors could clarify this point both in their response and in the manuscript.”

Our response:

We warmly thank the reviewer for emphasizing procedural clarification in experimental validation. Our revised manuscript now comprehensively details the methodology: For each target cyclopeptide sequence of length n , we systematically generated n distinct linear precursors by rotating the N- and C-termini positions across all possible cyclization sites. Taking cyclo-KCL as an example, we synthesized and cyclized all three linear permutation variants (KCL, CLK, and LKC). Similarly, for 7-residue cyclopeptide cyclo-APYCGSI, we prepared seven linear precursors (APYCGSI, PYCGSIA, YCGSIAP, CGSIAPY, GSIAPYC, SIAPYCG, IAPYCGS) and subjected each to cyclization under standardized conditions. Model accuracy was quantified as

the percentage of correctly predicted cyclization outcomes across all validated linear precursors. Specifically, in the cyclo-APYCGSI case, the model accurately predicted cyclization outcomes for six precursors but failed to anticipate cyclization for CGSIAPY, yielding a sequence-level accuracy of 86% (6/7). Across the entire validation set comprising 7 cyclopeptides and 42 precursors, the aggregate accuracy reached 83% (35 correct predictions). This methodology has now been explicitly detailed in “**Experimental Validation of ML Model**” section in the **manuscript**. We believe this clarification fully resolves the methodological ambiguity noted by the reviewer.

Before:

To assess the predictive accuracy of our ML model, we first evaluated its performance on three randomly generated cyclic peptide sequences: two 7-residue sequences, cyclo-APYCGSI and cyclo-YLHIEGV, and a 3-residue sequence, cyclo-KCL (Fig. 4a–c). Each peptide was subjected to cyclization under standard conditions, and the reaction mixtures were analyzed via LC-MS to verify the formation of the desired cyclic products. The experimental results demonstrated strong agreement with the model’s predictions, with cyclization accuracies of 86% for both cyclo-APYCGSI and cyclo-YLHIEGV, and 100% for cyclo-KCL. These findings underscore the model’s robustness in identifying viable cyclization sites.

*To further validate the model’s practical utility, we extended our investigation to four biologically active cyclic peptides derived from *Pseudostellariae Radix*32, including Heterophyllin J (5-residue), Pseudostellarins D (7-residues), Pseudostellarins A (5-residue), and Pseudostellarins H (8-residue) (Fig. 4d–g). Comparative analysis between the predicted and experimentally observed cyclization efficiencies revealed a high degree of consistency, with prediction accuracies of 100% for Heterophyllin J, 71% for Pseudostellarins D, 80% for Pseudostellarins A, and 75% for Pseudostellarins H.*

After conducting 42 synthesis experiments across seven distinct cyclic peptide sequences, the aggregate predictive accuracy of the model reached 83% (Fig. 4h), demonstrating its reliability in guiding the selection of optimal cyclization sites for

peptide synthesis. This high success rate highlights the model's potential as a valuable tool for synthetic chemists, streamlining the design and optimization of cyclic peptides with minimal experimental trial and error.

After:

To assess the predictive performance of our ML model, we implemented a validation protocol wherein each candidate cyclic peptide sequence was systematically evaluated across all potential cyclization sites. For the three randomly generated sequences (**Fig. 4a–c**), two 7-residue peptides (cyclo-APYCGSI and cyclo-YLHIEGV) and one 3-residue peptide (cyclo-KCL), we synthesized every linear precursor by changing cyclization sites. Specifically, cyclo-KCL yielded three linear precursors (KCL, CLK, LKC), while each 7-residue peptide generated seven precursors. Following automated solid-phase synthesis using DAN resin under standard conditions with reaction outcomes verified via LC-MS detection of the desired cyclic products. The experimental results demonstrated strong agreement with the model's predictions, with cyclization accuracies of 86% for both cyclo-APYCGSI and cyclo-YLHIEGV, and 100% for cyclo-KCL. These findings underscore the model's robustness in identifying viable cyclization sites.

To further validate the model's practical utility, we extended our investigation to four biologically active cyclic peptides derived from *Pseudostellariae Radix*32, including Heterophyllin J (5-residue), Pseudostellarins D (7-residues), Pseudostellarins A (5-residue), and Pseudostellarins H (8-residue) (**Fig. 4d–g**). Comparative analysis between the predicted and experimentally observed cyclization efficiencies revealed a high degree of consistency, with prediction accuracies of 100% for Heterophyllin J, 71% for Pseudostellarins D, 80% for Pseudostellarins A, and 75% for Pseudostellarins H.

After conducting 42 synthesis experiments across seven distinct cyclic peptide sequences, 35 predictions were consistent with experimental results. The aggregate predictive accuracy of the model reached 83% (**Fig. 4h**), demonstrating its reliability in guiding the selection of optimal cyclization sites for peptide synthesis. This high success rate highlights the model's potential as a valuable tool for synthetic chemists, streamlining the design and optimization of cyclic peptides with minimal experimental

trial and error.

Comment 3:

“It was not clear in the initial training dataset if the training peptides were synthesised as all possible linear starts/stops as illustrated in results for Fig 4. Does 306 peptides mean 306 unique peptides with all their different permutations (so more than 306 sequences in the actual dataset) or 306 unique sequences (which are permutations of each other)? The larger peptides would be much more over-represented in training if the former is true.”

Our response:

We sincerely appreciate the reviewer’s astute inquiry regarding the structure of our training dataset, which highlights a critical methodological detail requiring clarification. To address this concern explicitly: The 306 entries in our training dataset exclusively comprise unique linear precursor sequences and their corresponding cyclization outcomes, not comprehensive sets of circular permutation variants for each peptide. Specifically, each entry represents a distinct linear sequence subjected to head-to-tail cyclization under standardized conditions. For example, the 4-residue peptide sequence FPGM constitutes a single dataset entry, reflecting solely the cyclization outcome of the linear precursor FPGM to cyclo-FPGM. It does not encompass permutation variants including PGMF, GMFP, and MFPG. Consequently, the dataset contains 306 independent sequence-cyclization pairs with no inherent sequence permutation relationships, ensuring balanced representation across peptide lengths (2-14 residues) without over-representation of larger peptides. This design principle is now explicitly stated in “**Dataset Building**” Section in the **manuscript**. We regret any ambiguity in the original description and thank the reviewer for prompting this essential clarification.

Before:

Given our focus on predicting cyclization feasibility, we systematically investigated a diverse set of 306 linear peptides ranging from 2 to 14 residues in length.

After:

Given our focus on predicting cyclization feasibility, we established a training dataset of 306 unique linear peptide sequences (2-14 residues), each corresponding to a single experimentally validated cyclization outcome. Crucially, each entry represents only the specified linear precursor. For example, “FPGM” exclusively denotes cyclization of linear FPGM to cyclo-FPGM, excluding permutation variants such as PGMF.

Comment_4:

“Figure 4 legend should specify what the symbols in colour 1/0 means, does this mean predicted to be cyclised, but not observed experimentally, or the other way around?”

Our response:

We commend the reviewer’s acute attention to graphical clarity, which identified critical ambiguity in Figure 4’s symbology. To clarify the symbolic notation in Figure 4, each cyclization site is annotated with a binary label “P/E”, where P denotes the predicted cyclization outcome (1 = feasible, 0 = infeasible) and E represents the experimental observation (1 = successful cyclization detected, 0 = no cyclic product observed). A concordant label (e.g., 1/1 or 0/0) indicates accurate prediction, whereas discordant pairs (e.g., 1/0 or 0/1) reflect prediction errors. Specifically, “1/0” signifies a false positive case where cyclization was predicted feasible but not experimentally viable. We have revised the figure by adding more explicit labeling to enhance the clarity of the results presented. This modification ensures that the figure conveys the information in a clearer and more straightforward manner. Also, this annotation system is now explicitly defined in the revised Figure 4 legend in **“Experimental Validation of ML Model”** Section in the **manuscript**. We regret any confusion caused by the original omission and appreciate the reviewer’s meticulous critique.

Before:

Fig. 4. Predicted cyclic peptide synthesis results agree with experimental validation. The amide bonds representing cyclization sites are marked in color. The corresponding verification results are presented in the form of ‘predicted result/experimental result’ within the colored boxes. If the box is surrounded by a gray dashed line, it indicates a discrepancy between the experimental and predicted results; otherwise, it indicates that the two results align, confirming the accuracy of the prediction. Cyclic peptides in a), b), and c) are cyclic peptides generated randomly. Cyclic peptides in d), e), f), and g) are cyclic peptides in *Pseudostellariae Radix*, which present tyrosinase inhibitory activity. The accuracy of each cyclic peptide and overall accuracy are represented in h).

After:

0/1 → P/E P: Predicted E: Experimental

Fig. 4. Predicted cyclic peptide synthesis results agree with experimental validation. The amide bonds representing cyclization sites are marked in color, with results presented in the form of ‘P/E’ within the colored boxes. ‘P’ indicates predicted cyclization (1 = feasible, 0 = infeasible), and ‘E’ indicates experimental outcome (1 = successful, 0 = not observed). Boxes with gray dashed lines indicate a discrepancy between the experimental and predicted results; otherwise, it indicates that the two results align, confirming the accuracy of the prediction. Cyclic peptides in a), b), and c) are cyclic peptides generated randomly. Cyclic peptides in d), e), f), and g) are cyclic peptides in *Pseudostellariae Radix*, which present tyrosinase inhibitory activity. The accuracy of each cyclic peptide and overall accuracy are represented in h).

Comment_5:

“Was the feature selection process repeated for each cross-validation fold, or for the 70% of the data in the training set, or for the whole dataset. The authors should clarify in their response and the manuscript. If the feature selection was performed in the whole dataset, that could introduce overfitting, it would have been more appropriate to perform it only in the training data or the cross-validation folds.”

Our response:

This incisive query regarding feature selection protocols is highly valued, because it is crucial for ensuring methodological rigor. To clarify our approach, the feature selection process was strictly nested within the 5-fold cross-validation framework to prevent data leakage and overfitting. This design principle is now clearly outlined in **Section 3.3** of the **Supplementary Information**.

Before:

To identify the features that significantly impact model performance from the 414-dimensional vector, a forward feature selection process was conducted utilizing the SVC. This process commenced with an empty feature set and iteratively added the most impactful feature (defined as the one enhancing model performance the most) at each iteration, guided by cross-validation accuracy. The feature selection process concluded when 40 features were identified, beyond which further improvements in accuracy were no longer observed. Overall, the model’s accuracy was enhanced from 0.71 to 0.86 (Supplementary Fig. 7).

After:

To identify features significantly impacting model performance from the 414-dimensional vector, we conducted forward feature selection using 5-fold cross-validation within the training data. This process commenced with an empty feature set and iteratively added the most impactful feature based on cross-validation accuracy improvement. Feature selection was terminated at 40 features where accuracy gains plateaued, yielding a final model accuracy of 0.86 from initial 0.71 (**Supplementary**

Fig. 7).

Comment_6:

“The process for generating the random sequences should be specified. It is also important that the authors report how similar these sequences are to the sequences in their training data. If they are too similar, then the authors cannot claim robust generalization.”

Our response:

We recognize the reviewer's valuable emphasis on clarifying our random sequence generation methodology and assessing similarity to training data. The three validation peptides (cyclo-APYCGSI, cyclo-YLHIEGV, and cyclo-KCL) were generated through a purely stochastic process: sequence lengths were randomly sampled from 2 to 14 residues with each amino acid independently selected from the 20 native amino acids using a uniform probability distribution. Comprehensive manual comparison against our training dataset confirmed no significant sequence similarity, with particular attention to the absence of ≥ 3 consecutive residue matches, ensuring these represent novel validation cases. The generation script is publicly available at <https://github.com/Yongboxiao/Selection-of-Cyclization-Sites>. This methodology robustly substantiates the model's generalization capability, and we have incorporated these clarifications in Section 4 “**Details of the validation experiment**” of the revised **Supplementary Information**. We appreciate this constructive feedback which strengthens our validation framework documentation.

Before:

The selected Pseudostellariae Radix cyclic peptides, along with the randomly generated cyclic peptides, were processed using the “Cyclic Peptides Disconnection Sequences Generated” section of the program available at <https://github.com/Yongboxiao/Selection-of-Cyclization-Sites> to disconnect them at all cyclization sites, thereby generating the corresponding linear peptide sequences for

*synthesis. These sequences were subsequently input into the “Prediction” section of the trained model for prediction. Experimental validation was performed to assess the accuracy of the model. The experimental process was consistent with the synthesis process in the dataset (see **Automated cyclic peptide synthesis**).*

After:

The three randomly generated cyclic peptides, cyclo-APYCGSI, cyclo-YLHIEGV, and cyclo-KCL, were created using “Random Cyclopeptide Generator”, our dedicated random sequence generator package. This tool employs a purely stochastic algorithm that: (1) randomly samples sequence lengths from 2 to 14 residues, and (2) independently selects each amino acid from the 20 native types with a uniform probability distribution. The “Random Cyclopeptide Generator” package has been uploaded to our GitHub repository (<https://github.com/Yongboxiao/Selection-of-Cyclization-Sites>).

These randomly generated peptides, along with the selected Pseudostellariae Radix-derived cyclic peptides, were processed using the “Cyclic Peptides Disconnection Sequences Generated” section of the program available in our GitHub repository to disconnect them at all cyclization sites, thereby generating the corresponding linear peptide sequences for synthesis. These sequences were subsequently input into the “Prediction” section of the trained model for prediction. Experimental validation was performed to assess the accuracy of the model. The experimental process was consistent with the synthesis process in the dataset (see **Automated cyclic peptide synthesis**).

Comment_7:

“The link to the code repository is not publicly available. The authors should make the code publicly available that is used to train the models and generate the sequences, to verify the claims made in the text.”

Our response:

The reviewer's advocacy for open science is profoundly respected. Our GitHub repository (<https://github.com/Yongboxiao/Selection-of-Cyclization-Sites>) has provided full public access since submission, containing all code and data resources for this study, encompassing seven core components: (1) the "Random Cyclopeptide Generator" algorithm for stochastic sequence generation for experimental validation, (2) the "Cyclic peptides disconnection sequences generated" module generating linear precursors, (3) complete code for model training (4) Prediction implementation scripts, (5) the dataset of 306 peptides for model training, (6) Sequence data from experimental validation studies, and (7) Sequence data of therapeutic cyclic peptides targeting various diseases. The repository's comprehensive README.md provides detailed execution protocols and dependency specifications.

Comment_8:

"The authors should make available the training and evaluation data in order to assess claims made."

Our response:

We enthusiastically embrace the reviewer's emphasis on data accessibility. All critical datasets are now fully accessible through three complementary channels: (1) The complete training dataset of 306 peptides and validation experimental results are provided in the Supplementary Information; (2) Machine-readable versions of all data are archived in our GitHub repository (<https://github.com/Yongboxiao/Selection-of-Cyclization-Sites>), including source CSV files for model training/evaluation.

Comment_9:

"The authors do not specify what proportion of the peptides were positive and which were not in the training dataset."

Our response:

The reviewer's focus on class balance is analytically appreciated. Our dataset comprises exactly 174 cyclization-viable sequences (labeled "1") and 132 non-viable sequences (labeled "0"), yielding a near-balanced distribution of 56.9% positive and 43.1% negative cases, indicating no significant bias toward either class. We thank the reviewer for prompting this important clarification, which underscores the statistical robustness of our training data for model generalization. The relevant content has been added and revised in the "**Dataset Building**" section of the **manuscript**.

Before:

Notably, shorter cyclic peptides (2- and 3-residue) exhibited distinct cyclization behaviors due to their constrained backbones, prompting us to prioritize these subsets for deeper analysis. The final stratified dataset comprised 45 two-residue, 53 three-residue, 120 four-to-six-residue, 69 seven-to-nine-residue, and 19 ten-residue and longer cyclic peptides. This stratified distribution facilitated robust model training and generalization.

After:

The dataset comprises 174 cyclization-viable sequences (labeled "1") and 132 non-viable sequences (labeled "0"), ensuring balanced representation without outcome bias. Notably, shorter cyclic peptides (2- and 3-residue) exhibited distinct cyclization behaviors due to their constrained backbones, prompting prioritized analysis of these subsets. Final stratification included 45 two-residue, 53 three-residue, 120 four-to-six-residue, 69 seven-to-nine-residue, and 19 ten-to-fourteen-residue peptides, enabling robust model generalization across diverse peptide architectures.

Reviewer #3

Comment_1:

"Although the distribution in peptide size is well presented, the authors should include the amino acid diversity score of the starting library. In other words, how

similar (or different) are the starting sequences in terms of amino acid similarity/identity and three-dimensional structure? How were the 306 peptides selected?”

Our response:

We commend the reviewer’s discerning analysis regarding our dataset’s sequence diversity design. The 306-peptide dataset was meticulously engineered to ensure comprehensive representation of all 20 native amino acids, with each type appearing in ≥ 20 distinct sequences to guarantee robust generalization capability. To maximize pattern discernment, we employed a directed variation strategy: seed sequences underwent residue addition, deletion, or substitution—as exemplified by TDDA (no cyclization) yielding successfully cyclized variants TDDAG and GTDDA. These minimal sequence alterations revealed critical structure-cyclability relationships more effectively than random sampling, while maintaining broad heterogeneity across the dataset. This approach achieved optimal generalization capacity without significant redundancy. The relevant content has been added and revised in the “**Methods**” section in the **manuscript**.

Before:

Rapid cyclic peptides dataset building

The main step of dataset construction is the synthesis of cyclic peptides. The first step is manual DAN resin synthesis. Forty milligrams of DAN resin (0.36 mmol/g loading) were used in all experiments in the dataset. The next step is automated synthesis. All cyclic peptides were synthesized on the automated instrument iChemAFS along with control software AutoXpert. The following stock solutions were used for cyclic peptide synthesis: Fmoc-protected amino acids including 20 native amino acids. Activating agent O-(7-Azabenzotriazol-1-yl)-N, N, N', N'-tetramethyluronium hexafluorophosphate (HATU) as a 0.38 M stock solution in DMF, DIEA (10% v/v in DMF), DIEA (1% v/v in DMF) and deprotection stock solution (40% piperidine and 1% HCOOH (v/v) in DMF). The automated process includes linear peptide synthesis at 90 °C, activation at room temperature (25 °C), and cyclization at 50 °C. The reaction

mixtures were characterized by LC-MS (for more details, see Supplementary Information).

After:

Rapid cyclic peptides dataset building

Comprising 306 unique linear peptides, the dataset was engineered to ensure comprehensive representation of all 20 native amino acids, with each type appearing in more than 20 distinct sequences for robust generalization. Through directed variation strategies, including residue addition (e.g., TDDA to TDDAG), deletion, and substitution, minimal sequence alterations revealed critical structure-cyclability relationships while maintaining broad heterogeneity without redundancy, as evidenced by TDDA (no cyclization) versus TDDAG/GTDDA (successful cyclization).

The main step of dataset construction is the synthesis of cyclic peptides. The first step is manual DAN resin synthesis. Forty milligrams of DAN resin (0.36 mmol/g loading) were used in all experiments in the dataset. The next step is automated synthesis. All cyclic peptides were synthesized on the automated instrument iChemAFS along with control software AutoXpert. The following stock solutions were used for cyclic peptide synthesis: Fmoc-protected amino acids including 20 native amino acids. Activating agent O-(7-Azabenzotriazol-1-yl)-N, N, N', N'-tetramethyluronium hexafluorophosphate (HATU) as a 0.38 M stock solution in DMF, DIEA (10% v/v in DMF), DIEA (1% v/v in DMF) and deprotection stock solution (40% piperidine and 1% HCOOH (v/v) in DMF). The automated process includes linear peptide synthesis at 90 °C, activation at room temperature (25 °C), and cyclization at 50 °C. The reaction mixtures were characterized by LC-MS (for more details, see Supplementary Information).

Comment_2:

“In assigning the binary classification, “1” signifies that the target molecular weight was observed and “0” if it was not. It is unclear if the target molecular weight refers exclusively to the parent ion and what degree of precision was considered for

successful identification. The authors should provide these details. Looking at the Supplementary Information, it appears that $[M+H]^+$ and $[M+Na]^+$ are being considered. The authors should also state if they attempted to validate successful cyclizations by checking for adducted masses or smaller fragments.”

Our response:

We gratefully acknowledge the reviewer’s rigorous assessment of our cyclization classification criteria. The binary labeling system requires definitive detection of $[M+H]^+$ or $[M+Na]^+$ parent ions via LC-MS for a “1” designation (successful cyclization), while “0” indicates absence of both target monomeric species. Although adducted masses (e.g., $[2M+H]^+$) and fragment ions were documented during analysis, these were exclusively treated as supplementary diagnostic indicators—never as primary evidence for cyclization success. This stringent classification protocol ensures unambiguous interpretation of experimental outcomes. We have updated **Section 2.5** of the **Supplementary Information** accordingly.

Before:

*The dataset was established through two primary processes: synthesis and detection of cyclic peptides. Synthesis was efficiently conducted using iChemAFS on a 40 mg scale of DAN resin (0.36 mmol/g loading) following standard protocols (refer to **Automated cyclic peptide synthesis**). For cyclic peptide detection, the mixture obtained after automated synthesis was used directly for LC-MS analysis without additional processing, with a sample injection volume of 1 μ L.*

After:

The dataset was established through two primary processes: synthesis and detection of cyclic peptides. Synthesis utilized iChemAFS with 40 mg DAN resin (0.36 mmol/g loading) following standard protocols (refer to **Automated Cyclic Peptide Synthesis**). Post-synthesis mixtures underwent direct LC-MS analysis without processing (1 μ L injection volume). **Critical cyclization success criteria mandated detection of $[M+H]^+$ or $[M+Na]^+$ parent ions for “1” classification, while absence of both monomeric species yielded “0” designation. Adducted masses (e.g., $[2M+H]^+$) and**

fragment ions were served as supplementary validation indicators instead of primary evidence for cyclization success.

Comment_3:

“One limitation of the training model is that it only accounts for the twenty native amino acids. It does not account for non-proteinogenic residues, which are frequently included in peptides to improve drug-like properties as well as binding affinity and target selectivity. It would be good to consider peptides harboring some of the most frequently encountered non-proteinogenic residues in the model.”

Our response:

We are grateful for the reviewer’s thoughtful insight regarding the model’s amino acid scope. The current framework intentionally focuses on the 20 native residues given their prevalence in naturally occurring therapeutic cyclopeptides. While non-proteinogenic residues offer valuable functional enhancements, their structural heterogeneity introduces computational complexities requiring specialized methodologies beyond this study's scope. Incorporating these residues constitutes a planned research initiative for next-phase development, necessitating expanded feature engineering and training paradigms. This work establishes a foundational framework for that forthcoming advancement, and we value the reviewer’s perspective in highlighting this critical progression pathway.

Comment_4:

“In the Supplementary Information, the ‘Found: [M+H]⁺’ for cyclo-YLHIEGV is incorrect. What is written does not match the provided MS spectra.”

Our response:

We profoundly thank the reviewer for this meticulous observation and apologize for the oversight. Upon re-examining the raw mass spectra, we confirm the reported [M+H]⁺ value for cyclo-YLHIEGV in the Supplementary Information was incorrect in

the annotation. The value has been corrected from “1066.60” to “1166.60” to accurately reflect the observed LC-MS data. To ensure data accuracy, we conducted a thorough re-examination of all LC-MS data in the SI. The revised Supplementary Information highlights this correction in yellow. We are grateful for the reviewer's rigorous scrutiny, which has significantly enhanced the reliability of our dataset.

Before:

LC-MS condition: Method B. MS (ESI) exact mass calcd. for C₆₆H₈₇N₉O₁₀, [M+H]⁺: 1166.67; [M+Na]⁺: 1188.65. Found: [M+H]⁺: 1066.60.

After:

LC-MS condition: Method B. MS (ESI) exact mass calcd. for C₆₆H₈₇N₉O₁₀, [M+H]⁺: 1166.67; [M+Na]⁺: 1188.65. Found: [M+H]⁺: 1166.60.

Comment_5:

“In the Supplementary Information, some of the peptides form [2M+Na]⁺ in addition to [M+H]⁺. As outlined in the successful cyclization criteria, this would still be considered successful because it made some of the correct product in addition to the dimer. Can the model predict oligomerization based on the current training data? Aside from cyclization, the risk of oligomerization would be an important parameter for peptide medicinal chemists designing sequences.”

Our response:

We are grateful for the reviewer's perceptive comment regarding the observation of oligomeric species like [2M+Na]⁺ alongside monomeric ions in the Supplementary Information and the relevance of oligomerization prediction for peptide design. Our model successfully identifies viable cyclization, including cases where [2M+Na]⁺ coexists with the monomeric product, consistent with our established cyclization criteria. However, explicit prediction of oligomerization propensity falls beyond the model's current capabilities due to fundamental limitations in the training data design. Empirical studies using DAN-resin-based automated synthesis indicate oligomerization

occurs predominantly in 3-4 residue peptides and exhibits significant, yet poorly characterized, sequence-dependent behavior (*ACS Comb. Sci.* **2013**, *15*, 120–129; *Amino Acids* **2015**, *47*, 1495–1505). This issue pertains to reaction pathways that compete with cyclization, the mechanisms of which cannot be deduced from sequence information alone. The development of more efficient methodologies may represent a promising approach to address this challenge. We fully concur with the reviewer's insightful perspective on the critical importance of oligomerization prediction for drug development. With the accumulation of additional relevant data in the future, we aim to build upon our current model framework to further develop predictive capabilities for such outcomes. Once again, we sincerely appreciate the reviewer's forward-thinking and highly valuable suggestions.

We have carefully enhanced the manuscript based on your constructive suggestions. All modifications in the revised manuscript and supplementary information are marked with yellow highlights and strengthen the work while preserving its content or structure. We sincerely thank you for your time and expertise in reviewing our work, and hope our responses and revisions meet your approval.

Sincerely,

Chengxi Li, Ph.D.

Assistant Professor,

College of Chemical and Biological Engineering, ZJU

ZJU-Hangzhou Global Scientific and Technological Innovation Center, ZJU

Response to Reviewers' Comments

Dear Reviewers,

We sincerely thank you for your rigorous evaluation and insightful critiques of our manuscript (ID: NCOMMS-25-33268A). Your expertise and meticulous attention to detail have significantly contributed to enhancing the scientific quality of our work.

In accordance with your valuable recommendations, we have diligently undertaken extensive revisions to the manuscript. Each of your comments has been addressed through substantive modifications in the main text and supplementary materials. Below, we provide a point-by-point response to each of your comments. For ease of reference, all reviewer comments are reproduced in italics, and revised content has been explicitly highlighted in blue within the updated documents.

Reviewer #1

Comment_1:

“in its current state the experimental validation is insufficient (only 7 peptides are used as a hold out test set). Moreover, given the enrichment in cyclization sites in the evaluation set (there are 32 positive cyclization sites against 10 negative ones), the expected model accuracy for a model that always predicts “1” would be 76.2% with CycloPepper showing an accuracy of 83%, is only 8% above.”

Our response:

We are grateful to the reviewer for highlighting these important issues. Some of the sequences used in the experimental validation were randomly generated by our program, and we had not initially noticed the imbalance in the validation dataset. To expand the validation set and achieve a balanced ratio of positive and negative cyclization sites, we randomly generated an additional batch of sequences using the same program and manually selected some sequences with a higher proportion of negatives for experimental validation. The experimental validation dataset now consists of 12 peptides with 74 sequences, including 39 positive cyclization sites, which account for 24.2% of the training set. We have updated the corresponding content in “**Experimental Validation of ML Model**” section of the **manuscript** and revised the associated figure accordingly.

Before:

To assess the predictive performance of our ML model, we implemented a validation protocol wherein each candidate cyclic peptide sequence was systematically evaluated across all potential cyclization sites. For the three randomly generated sequences (Fig. 4a–c)—two 7-residue peptides (cyclo-APYCGSI and cyclo-YLHIEGV) and one 3-residue peptide (cyclo-KCL)—we synthesized every linear precursor by changing cyclization sites. Specifically, cyclo-KCL yielded three linear precursors (KCL, CLK, LKC), while each 7-residue peptide generated seven precursors. Following automated solid-phase synthesis using DAN resin under standard conditions with reaction outcomes verified via LC-MS detection of the desired cyclic products. The experimental results demonstrated strong agreement with the model's predictions, with cyclization accuracies of 86% for both cyclo-APYCGSI and cyclo-YLHIEGV, and 100% for cyclo-KCL. These findings underscore the model's robustness in identifying viable cyclization sites.

After:

To assess the predictive performance of our ML model, we implemented a validation protocol wherein each candidate cyclic peptide sequence was systematically evaluated across all potential cyclization sites. For the 8 randomly generated sequences (Fig. 4a–h): five 7-residue sequences, cyclo-APYCGSI, cyclo-YLHIEGV, cyclo-QSVQERS, cyclo-QEYQHEK, cyclo-WMLWEPF, a 6-residue sequence, cyclo-MMVMPY, a 5-residue sequence, cyclo-HMHPW, and a 3-residue sequence, cyclo-KCL, we synthesized every linear precursor by changing cyclization sites. Specifically, cyclo-KCL yielded three linear precursors (KCL, CLK, LKC), while each 7-residue peptide generated seven precursors. Following automated solid-phase synthesis using DAN resin under standard conditions with reaction outcomes verified via LC-MS detection of the desired cyclic products. The experimental results demonstrated strong agreement with the model's predictions, with cyclization accuracies of 71% for both cyclo-QEYQHEK and cyclo-WMLWEPF, 86% for cyclo-QSVQERS, and 100% for cyclo-APYCGSI, cyclo-YLHIEGV, cyclo-KCL, cyclo-HMHPW and cyclo-MMVMPY. These findings underscore the model's robustness in identifying viable cyclization sites.

Comment 2:

“the model seems to be always predicting peptide cyclization sites where the N belongs to Pro to be negative, even in some cases where it is positive. I would

recommend an ablation study to see whether the combination of features used in the study is necessary or positive, or whether the one-hot encoding is sufficient to make the predictions.”

Our response:

We sincerely thank the reviewer for the valuable suggestion. Following the recommendation, we have conducted an ablation study on all selected features to further assess their contribution. In this study, each feature was removed from the selected feature set in turn, and five-fold cross-validation was performed. The removal of any single feature resulted in a decrease in accuracy, with the removal of the N-Pro feature leading to a 2.5% drop in accuracy, indicating that this feature contributes positively to the model’s predictive performance. The corresponding description and figure have been added to **Section 3.3** in **Supporting Information**.

Before:

There is no relevant content in the Supplementary Information.

After:

The following is the newly added content:

To further evaluate the contribution of the selected features, an ablation study was performed. In this procedure, each feature was removed individually from the final feature set, and the models were retrained and evaluated using 5-fold cross-validation. Model performance was again measured by accuracy. The results showed that the removal of any single feature led to a decrease in cross-validation accuracy, indicating that all selected features contributed positively to the predictive performance of the models (**Supplementary Fig. 7**).

Supplementary Fig. 7. Cross-validation Accuracy in Feature Ablation Study for SVC.

Comment_3:

“Reading Comment_9, the proportion of labels in the training dataset is 56.9%, which is our closest estimate to the expected probability of a randomly sampled peptide to be cyclizable. However, the expected probability looking only at the evaluation set is 76.2%. This highlights the problem with the sample size in the evaluation set, which does not allow for even getting appropriate estimators for the expected probability of cyclization sites.”

Our response:

We sincerely thank the reviewer for the constructive comments. In light of the issues raised, we have supplemented the experimental validation dataset and adjusted the probability accordingly, as described in Comment 1. Now the expected probability of a randomly sampled peptide to be cyclizable in the evaluation set is 52.7% (39 feasible cyclization-sites), which is close to the proportion in the training dataset. We have updated the corresponding content in the **“Experimental Validation of ML Model”** section of the **manuscript**.

Before:

After conducting 42 synthesis experiments across seven distinct cyclic peptide sequences, 35 predictions were consistent with experimental results.

After:

In the entire experimental validation set, 39 sequences were cyclizable, while 35 were non-cyclizable, providing a nearly balanced dataset for evaluating model performance. After conducting 74 synthesis experiments across seven distinct cyclic peptide sequences, 64 predictions were consistent with experimental results.

Comment_4:

“Finally, after reviewing the code, there is an additional problem with the approach, the MinMaxScaler that is applied to X_train in the code block that starts with “#Concatentation” should not be fitted to the whole training dataset, it should be applied to each training fold individually as it is part of model building and feature selection.”

Our response:

We thank the reviewer for pointing out this critical issue. Proper use of the MinMaxScaler is indeed essential, as applying it to the entire dataset may cause data leakage and impair

generalization. We have corrected this error, re-run model selection, training, and prediction, . After this correction, we also slightly adjusted the model selection strategy by applying feature selection uniformly across all candidate models before performance comparison. This led to minor changes in the results, which have likewise been updated in both the **Manuscript** and the **Supplementary Information**.

Before:

The original 716-dimensional vector was normalized using the MinMaxScaler function, resulting in the elimination of all zero dimensions and a final dimensionality of 414. This 414-dimensional vector was then utilized as input for twelve different machine learning models. Through five-fold cross-validation, the Support Vector Classifier (SVC) displayed the highest average accuracy, achieving a score of 0.71 (**Supplementary Fig. 6**).

Supplementary Fig. 6. Average accuracy of 12 models using 414-dimensional features.

To identify features significantly impacting model performance from the 414-dimensional vector, we conducted forward feature selection using 5-fold cross-validation within the training data. This process commenced with an empty feature set and iteratively added the most impactful feature based on cross-validation accuracy improvement. Feature selection was terminated at 40 features where accuracy gains plateaued, yielding a final model accuracy of 0.86 from initial 0.71 (**Supplementary Fig. 7**).

Supplementary Fig. 7. Average accuracy during epochs in forward feature selection.

After:

The original 716-dimensional vector undergoes zero-dimensional elimination, resulting in a final dimension of 414. This 414-dimensional vector was then utilized as input for twelve different machine learning models.

To identify features significantly impacting model performance from the 414-dimensional vector, we conducted forward feature selection using 5-fold cross-validation within the training data. This process commenced with an empty feature set and iteratively added the most impactful feature based on cross-validation accuracy improvement (including MinMaxScalar in each fold). If no improvement in accuracy was observed for five consecutive rounds, the selection process was terminated. Comparing the five-fold cross-validation results of 12 models after feature selection, the Support Vector Classifier (SVC) displayed the highest average accuracy, achieving a score of 0.84. The feature selection process concluded when 31 features were identified. Overall, the model's accuracy was enhanced from 0.68 to 0.84 (Supplementary Fig. 6).

Supplementary Fig. 6. Average accuracy during epochs in forward feature selection for SVC.

Comment_5:

“We continue recommending that the standard deviation or standard error of the mean across the 5 folds is included when reporting model performance, not only the average.”

Our response:

We express our sincere appreciation for the suggestions provided by the reviewer. We have reported the model standard deviations in the “**ML Model Selection and Performance Evaluation**” section in the **manuscript**.

Before:

Model performance was evaluated using five-fold cross-validation to ensure statistical reliability. The GaussianNB classifier exhibited the lowest average accuracy (0.61), while seven models yielded accuracies within the range of 0.6–0.7. Three models demonstrated improved performance, achieving accuracies between 0.7 and 0.8. Notably, two models surpassed an accuracy threshold of 0.8, with the SVC emerging as the top-performing algorithm, delivering a mean accuracy of 0.86 (Fig. 3). This superior predictive capability highlights the effectiveness of SVC in capturing the underlying structure-property relationships within our dataset. Given its robust performance, the SVC model was selected for subsequent experimental validation and predictive applications.

After:

Model performance was evaluated using five-fold cross-validation to ensure statistical reliability. The GaussianNB classifier displayed the lowest performance (0.74 ± 0.05). Several models, including RF (0.79 ± 0.06), LR (0.81 ± 0.07), and XGB (0.76 ± 0.06), achieved moderate accuracies within the 0.70-0.80 range. Enhanced performance with relatively low variance was observed for GB (0.82 ± 0.04), Adaboost (0.82 ± 0.05), and the stacking (0.82 ± 0.06). Notably, KNN (0.83 ± 0.06) and SVC (0.84 ± 0.06) delivered the highest mean accuracies, with SVC providing the optimal balance between accuracy and stability (Fig. 3). Given its reliable performance, the SVC model was selected for subsequent experimental validation and predictive applications.

Comment_6:

“Similarly, figure 3 should also display error bars representing that deviation.”

Our response:

We sincerely thank the reviewer for the valuable suggestion. We have added error bars to **Figure 3** in the **manuscript**.

Fig.3. Overview of machine learning models. Input features include one-hot encoding sequences, AAC, DPC, and domain-specific hand-crafted features. The principle of each encoding is explained in the figure. Model selection was according to the average accuracy values of the models. The GaussianNB model presented the lowest value: 0.61. The SVC model presented the highest value: 0.86.

After:

Fig.3. Overview of machine learning models. Input features include one-hot encoding sequences, AAC, DPC, and domain-specific hand-crafted features. The principle of each encoding is explained in the figure. Model selection was according to the average accuracy values of the models. The GaussianNB model presented the lowest value: 0.74. The SVC model presented the highest value: 0.84.

Comment_7:

*“The similarity between the sequences in training and testing should be reported (see [2]) to give a sense of how generalizable the approach really is. This includes both random sequences and the sequences from *Pseudostellariae radix*.”*

Our response:

We sincerely thank the reviewer for the constructive comments and insightful suggestions. In accordance with the DOME guidelines, we have calculated the percent identity as the sequence similarity between the training set and all sequences used for experimental validation, including both random sequences and those derived from *Pseudostellariae radix*. The corresponding results have been added to the “**Similarity Evaluation**” section of the manuscript and Section 4.2 of the Supporting Information.

Before:

Featurization

Twelve ML model architectures were implemented in the Python programming environment, version 3.7.1. Fivefold cross-validation was used for a random split of 70% training and 30% held-out testing datasets. ML algorithms were mostly imported from the

module “scikit-learn”.

After:

Featurization

Twelve ML model architectures were implemented in the Python programming environment, version 3.7.1. Fivefold cross-validation was used for a random split of 70% training and 30% held-out testing datasets. ML algorithms were mostly imported from the module “scikit-learn”.

Similarity Evaluation

Following the DOME recommendations for supervised machine learning validation in biology, we measured sequence similarity as percent identity, quantifying the independence between the training set and experimental validation set by computing global pairwise identity between each experimental amino-acid sequence and all training sequences on the linearized peptides. Since cyclic peptides can be represented with different starting positions when linearized, we performed a rotation-invariant audit for high-similarity pairs (initial identity \geq 60%) (for more details, see Supporting Information Section 4.2).

Comment_8:

“In Figure 4, there is a typo in the species name which should be italicised and the second name in lower case”

Our response:

We sincerely thank the reviewer for pointing out this error. The typographical mistake in **Figure 4**, as well as in the **manuscript** and **Supporting Information**, has now been corrected.

Before:

0 / 1 → P/E P: Predicted E: Experimental

Fig. 4. Predicted cyclic peptide synthesis results agree with experimental validation. The amide bonds representing cyclization sites are marked in color, with results presented in the form of 'P/E' within the colored boxes. 'P' indicates predicted cyclization (1 = feasible, 0 = infeasible), and 'E' indicates experimental outcome (1 = successful, 0 = not observed). Boxes with gray dashed lines indicate a discrepancy between the experimental and predicted results; otherwise, it indicates that the two results align, confirming the accuracy of the prediction. Cyclic peptides in a), b), and c) are cyclic peptides generated randomly. Cyclic peptides in d), e), f), and g) are cyclic peptides in *Pseudostellariae Radix*, which present tyrosinase inhibitory activity. The accuracy of each cyclic peptide and overall accuracy are represented in h).

After:

Fig. 4. Predicted cyclic peptide synthesis results agree with experimental validation. The amide bonds representing cyclization sites are marked in color, with results presented in the form of ‘P/E’ within the colored boxes. ‘P’ indicates predicted cyclization (1 = feasible, 0 = infeasible), and ‘E’ indicates experimental outcome (1 = successful, 0 = not observed). The corresponding verification results are presented in the form of ‘predicted result/experimental result’ within the colored boxes. If the box is surrounded by a gray dashed line, it indicates a discrepancy between the experimental and predicted results; otherwise, it indicates that the two results align, confirming the accuracy of the prediction. Cyclic peptides in a) to h) are cyclic peptides generated randomly. Cyclic peptides in i), j), k), and l) are cyclic peptides in *Pseudostellariae radix*, which present tyrosinase inhibitory activity. The accuracy of each cyclic peptide and the overall accuracy are represented in m).

Comment 9:

“Authors should include further details on what the seed sequences are. Additionally, if the directed variation strategies were performed according to an algorithm it should be clearly stated and the text should direct to the algorithm in the supplementary; alternatively, it should be clearly stated that this process was manually

performed.”

Our response:

We appreciate the reviewer’s suggestion for further clarification. A total of 22 seed sequences (SFEGMPN, MSFEGMP, YNSFPGM, NGTKGDY, DGAFQAY, LDPGTFI, CGDYGTK, HTLGTID, FEGMPNS, RGFACAY, WRDGGTDDA, QVEDGTHYK, RHDADGGTD, SEAVDGTAK, MKWPFPGPTL, NSGGTDDAD, CMILDPGTF, KRDPGTFIL, HVIGPSYFG, HCAYRGFAA, TYRGFAACA, ICWPFPGPTL) were initially generated by a programmatic approach to ensure representation of all 20 proteinogenic amino acids. From this pool, subsequences were manually selected to maximize diversity in amino acid composition while retaining compatibility with the directed variation strategies, ultimately yielding a library of 306 sequences (Supplementary Table 2). This design ensured that the resulting variants captured the full range of structural diversity and cyclization potential, thereby rendering the dataset comprehensive and robust. The “**Methods**” section of the **manuscript** and **Supporting Information Section 2.5** have been updated accordingly.

Before:

Rapid cyclic peptides dataset building

Comprising 306 unique linear peptides, the dataset was engineered to ensure comprehensive representation of all 20 native amino acids, with each type appearing in more than 20 distinct sequences for robust generalization. Through directed variation strategies, including residue addition (e.g., TDDA to TDDAG), deletion, and substitution, minimal sequence alterations revealed critical structure-cyclability relationships while maintaining broad heterogeneity without redundancy, as evidenced by TDDA (no cyclization) versus TDDAG/GTDDA (successful cyclization).

After:

Rapid cyclic peptides dataset building

Comprising 306 unique linear peptides, the dataset was engineered to ensure comprehensive representation of all 20 native amino acids, with each type appearing in more than 20 distinct sequences for robust generalization. **The seed sequences were randomly generated using a programmatic approach to ensure the inclusion of all 20 native amino acids. From this pool, specific subsequences were selected for the directed variation strategies, including residue addition (e.g., TDDA to TDDAG), deletion, and substitution. These modifications were manually performed (Supporting Information section 2.5). Minimal sequence alterations revealed critical structure-cyclability relationships while maintaining**

broad heterogeneity without redundancy, as evidenced by TDDA (no cyclization) versus TDDAG/GTDDA (successful cyclization).

We have endeavored to enhance the manuscript and have implemented several modifications in the revised paper, which are marked with highlights. These changes do not alter the content or structure of the paper. We sincerely appreciate the diligent efforts of the Editors/Reviewers and hope that the corrections meet with your approval. We trust that the revised manuscript is now suitable for publication in *Nature Communications*.

I look forward to hearing from you.

Sincerely,

Chengxi Li, Ph.D.

Assistant Professor,

College of Chemical and Biological Engineering, ZJU

ZJU-Hangzhou Global Scientific and Technological Innovation Center, ZJU

Dear Reviewers,

We sincerely thank you for your valuable comments on our revised manuscript (ID: NCOMMS-25-33268B). We are also grateful for your positive evaluation of the revisions made in the previous round. In this response, we have addressed each of the remaining comments point by point. All corresponding changes have been highlighted in the revised **Manuscript** and **Supporting Information** for your convenience.

Reviewer #1

Comment_1:

“There is still the issue of the dataset and evaluation sets being too small for the strong claim of a "robust generalization", we do not see anything in the paper to justify that claim.”

Our response:

We thank the reviewer for this valuable comment. We fully acknowledge that the limited size of our dataset constrains the strength of any claim regarding the model’s generalization capability. Accordingly, we have revised the manuscript to avoid using strong assertions such as “robust generalization” and have appropriately toned down related statements throughout both the **main text** and the **Supporting Information**. All corresponding modifications have been highlighted in the revised version.

Comment_2:

“With regards to the analysis, it is clear from the standard deviation that there is no significant effect of the choice of learning algorithm (all of them are within 1 std), the way in which the results are expressed seems to suggest the opposite. It is reasonable to choose the model with the highest average performance.”

Our response:

We sincerely thank the reviewer for this insightful comment regarding the interpretation of model performance. We fully agree that the differences among the learning algorithms are not statistically significant, as all results lie within one standard deviation. Following this suggestion, we have revised the relevant descriptions in both the **main text** and the **Supplementary Information** to eliminate any potential ambiguity. The text now clearly states that, while the performance differences are minor, the model with the highest **average**

performance was selected as the final choice.

Before:

“Model performance was evaluated using five-fold cross-validation to ensure statistical reliability. The GaussianNB classifier displayed the lowest performance (0.74 ± 0.05). Several models, including RF (0.79 ± 0.06), LR (0.81 ± 0.07), and XGB (0.76 ± 0.06), achieved moderate accuracies within the 0.70-0.80 range. Enhanced performance with relatively low variance was observed for GB (0.82 ± 0.04), Adaboost (0.82 ± 0.05), and the stacking (0.82 ± 0.06). Notably, KNN (0.83 ± 0.06) and SVC (0.84 ± 0.06) delivered the highest mean accuracies, with SVC providing the optimal balance between accuracy and stability (Fig. 3). Given its robust performance, the SVC model was selected for subsequent experimental validation and predictive applications.”

After:

“Model performance was evaluated using five-fold cross-validation to ensure statistical reliability. The results indicated that the differences among the learning algorithms were not statistically significant, as all performances fell within one standard deviation. The GaussianNB classifier displayed the lowest performance (0.74 ± 0.05), while several models, including RF (0.79 ± 0.06), LR (0.81 ± 0.07), and XGB (0.76 ± 0.06), achieved moderate accuracies in the 0.70–0.80 range. Slightly higher accuracies were obtained for GB (0.82 ± 0.04), AdaBoost (0.82 ± 0.05), and the stacking model (0.82 ± 0.06). Notably, KNN (0.83 ± 0.06) and SVC (0.84 ± 0.06) achieved the highest mean accuracies. Given the close performance and comparable variance, the SVC model with the highest mean accuracy was selected for subsequent experimental validation and predictive applications (Fig. 3).”

Comment_3:

“Fivefold cross validation cannot be used with a random split of 70% training and 30% test; fivefold means there are five equally sized folds each of them with 20% of the data ($100/5=20$), training is performed in 4 folds and testing on the fifth, in that way all folds are used for testing once. Did you perform fivefold cross-validation or repeated the experiment five times with 70/30 random split? Both things are mutually exclusive.”

Our response:

We sincerely thank the reviewer for this careful observation. It is absolutely correct that fivefold cross-validation and a 70/30 random split are mutually exclusive procedures. In model selection part of our study, we performed standard fivefold cross-validation, where the dataset

was divided into five equal parts, each used once for testing while the remaining four parts were used for training. The statement in the manuscript was an inadvertent mistake in wording. We have corrected the description in the **manuscript** to accurately reflect this procedure.

Before:

Featurization

Twelve ML model architectures were implemented in the Python programming environment, version 3.7.1. Fivefold cross-validation was used for a random split of 70% training, and 30% held-out testing datasets. ML algorithms were mostly imported from the module “scikit-learn”.

After:

Featurization

*“Twelve ML model architectures were implemented in the Python programming environment, version 3.7.1. Fivefold cross-validation was used for **evaluating model performance**. ML algorithms were mostly imported from the module “scikit-learn”.*”

We have endeavored to improve the manuscript and have implemented several revisions in the revised version, which are highlighted for clarity. These changes do not affect the content or structure of the paper. We sincerely appreciate the constructive efforts of the Reviewers and hope that the corrections meet with your approval. We trust that the revised manuscript is now suitable for publication in *Nature Communications*.

We look forward to your response.

Sincerely,

Chengxi Li, Ph.D.

Assistant Professor,

College of Chemical and Biological Engineering, ZJU

ZJU-Hangzhou Global Scientific and Technological Innovation Center, ZJU